# Mesoporous Highly-Deformable Composite Polymer for a Gapless Triboelectric Nanogenerator via a One-Step Metal Oxidation Process

**DOI:** 10.3390/mi9120656

**Published:** 2018-12-11

**Authors:** Hee Jae Hwang, Younghoon Lee, Choongyeop Lee, Youngsuk Nam, Jinhyoung Park, Dukhyun Choi, Dongseob Kim

**Affiliations:** 1Department of Mechanical Engineering, Kyung Hee University, 1732 Deogyeong-daero, Giheung-gu, Yongin-Si, Gyeonggi-do 446701, Korea; hjhwang@khu.ac.kr (H.J.H.); younghoon@snu.ac.kr (Y.L.); cylee@khu.ac.kr (C.L.); ysnam1@khu.ac.kr (Y.N.); 2Construction Equipment R&D Group, Korea Institute of Industrial Technology (KITECH), 288-1, Daehak-ri, Hayang-eup, Gyeongsan-si, Gyeongsangbuk-do 712091, Korea; jh.park@kitech.re.kr; 3Aircraft System Technology Group, Korea Institute of Industrial Technology (KITECH), 57, Yangho-gil, Yeongcheon-si, Gyeongbuk-do 38822, Korea

**Keywords:** mesoporous composite polymer, metal oxidation, gapless, triboelectric nanogenerator, high deformability

## Abstract

The oxidation of metal microparticles (MPs) in a polymer film yields a mesoporous highly-deformable composite polymer for enhancing performance and creating a gapless structure of triboelectric nanogenerators (TENGs). This is a one-step scalable synthesis for developing large-scale, cost-effective, and light-weight mesoporous polymer composites. We demonstrate mesoporous aluminum oxide (Al_2_O_3_) polydimethylsiloxane (PDMS) composites with a nano-flake structure on the surface of Al_2_O_3_ MPs in pores. The porosity of mesoporous Al_2_O_3_-PDMS films reaches 71.35% as the concentration of Al MPs increases to 15%. As a result, the film capacitance is enhanced 1.8 times, and TENG output performance is 6.67-times greater at 33.3 kPa and 4 Hz. The pressure sensitivity of 6.71 V/kPa and 0.18 μA/kPa is determined under the pressure range of 5.5–33.3 kPa. Based on these structures, we apply mesoporous Al_2_O_3_-PDMS film to a gapless TENG structure and obtain a linear pressure sensitivity of 1.00 V/kPa and 0.02 μA/kPa, respectively. Finally, we demonstrate self-powered safety cushion sensors for monitoring human sitting position by using gapless TENGs, which are developed with a large-scale and highly-deformable mesoporous Al_2_O_3_-PDMS film with dimensions of 6 × 5 pixels (33 × 27 cm^2^).

## 1. Introduction

Mechanical energy is the most common energy in our surroundings, and it can be converted into electricity anytime and anywhere. Triboelectric nanogenerators (TENGs), which are power-generating devices introduced in 2012, have been proven as cost effective, simple, able to cover large areas, durable, and efficient for energy harvesting [1,2,3]. Due to their simple mechanism, which utilizes repeating cycles of contact and separation between two materials, TENGs have demonstrated promising capability in scavenging energy and as variable sensors of mechanical energy (vibration, rotating, etc.), ocean waves, fluids, and human activities. Further, TENGs can generate electric signals without relying on external power sources, thus enabling the development of self-powered Internet of Things (IoT) sensors [4,5].

With a contact-separation structure, the surface charge density of the tribo-material is the key to achieving a high TENG output performance. To increase the surface charge, many fabrications and systems have been introduced in previous studies such as micro-patterned arrays, porous structures, multilayer alignment, ion injections, ground systems, mixing of high dielectric constant materials, and charging pumps [6,7,8,9,10,11,12]. Among them, it is still requested to develop a new porous fabrication technique to overcome the limits of fabricating porous structures. The reasons for studying porous structures is not only limited to increasing the output performance, but also applies to life because foam materials apply to many fields such as matrix structures, seat cushions, and shoe inserts. In previous studies, the fabrication of porous structures has generally consisted of two steps, suspensions and removal, which include techniques such as solvent casting/particle leaching and melting molds. However, they have more two steps in their fabrication methods and require more time to fabricate than TENG samples. Furthermore, these fabrications are hard to scale up and suffer from a lack of control over end product porosity [7,13,14,15].

In this work, we report a facile one-step metal oxidation process for creating mesoporous polymer films with a highly deformable composite structure to improve the capacitance significantly, thus resulting in enhancing the output performance of TENGs. The highly-deformable composite polymers are fabricated by casting a mixture of polymer mixed metal microparticle (MP) solution, like polydimethylsiloxane (PDMS), deionized (DI) water, and aluminum (Al) MPs, followed by evaporating water in an 80 °C oven, where the concentrations of Al MPs are in the range of 0–15 wt%. Interestingly, it is found that the surface of Al MPs is oxidized, and nano-flake structures are formed. As a result, we can obtain a mesoporous PDMS polymer integrated with aluminum oxide (Al_2_O_3_) MPs with the nano-flake surfaces. Based on the mesoporous highly-deformable PDMS composite polymer, the TENG output shows significant enhancement, as compared to a non-porous PDMS. The mesoporous PDMS composite with the weight fraction of 15 wt% exhibits the highest voltage of 259.58 V and current of 7.16 µA under 33.3 kPa and at 4 Hz. For a practical application, we demonstrate a self-powered safety cushion sensor with optimized TENG output performance using a low-cost, simple synthesis and that is light-weight to cover a large area of 33 × 27 cm^2^ (6 × 5 pixels).

## 2. Materials and Methods

### 2.1. The Fabrication Process of the Mesoporous Al_2_O_3_-PDMS

A mesoporous Al_2_O_3_-PDMS structure was fabricated by a one-step method, as shown in Figure 1a. In a previous study, porous PDMS embedded powder was made by a two-step mixing and etching process. However, in the present study, we introduce a one-step method for the porous structure. The PDMS was prepared as a mixture of base resin and curing agent (Sylgard 184 A:Sylgard 184 B, Dow Corning Co., Midland, MI, USA) at a weight ratio of 10:1. Al MPs (1 μm–10 μm), and DI water was added to the solution at weight ratios from 5%–15% and 50%. After degassing under vacuum for 20 min, the solution was laminated on the Teflon mold at a thickness of 3 mm and cured at 80 °C for six hours. The mesoporous Al_2_O_3_-PDMS was then peeled off the Teflon mold. Al tape (70 μm) was attached to the bottom of mesoporous Al_2_O_3_-PDMS by the electrode layer. 

### 2.2. Fabrication and Output Measurement of the TENGs

To design the vertical contact-separation mode TENG, the top electrode was prepared with a 3 × 3 cm^2^ 30-µm-thick conductive graphene tape (DASAN Solueta, SSC30) and mesoporous Al_2_O_3_-PDMS, with a thickness of 3 mm. The conductive graphene tape and mesoporous Al_2_O_3_-PDMS were attached to the acryl with double-sided tape.

To apply the vertical contact-separation mode, we used a pushing tester (JPT-110) ranging from 5 N–30 N of force. To measure TENG output performance, we used an oscilloscope (Tektronix MDO 3052, Beaverton, OR, USA) probe with 40 MΩ and a preamplifier (Stanford Research Systems, SR570, Sunnyvale, CA, USA) connected to the MDO 3052. TENG performance was measured at a working frequency of 4 Hz, and the gap distance was 4 mm between the two materials. 

### 2.3. Analysis of the Al_2_O_3_-PDMS TENG

The surface morphology, cross-section, energy dispersive spectroscopic (EDS) mapping, and line scan characteristics of the PDMS layer and mesoporous Al_2_O_3_-PDMS were examined using high-resolution field emitting scanning electron microscopy (HR FE-SEM, Leo supra 55, Carl Zeiss, Stockholm, Sweden). The porosity was measured by changes of volume and mass. Compressive strength was tested using a Universal Testing System (Instron, MA, USA) following ASTM D 695. Capacitance measurements of the mesoporous Al_2_O_3_-PDMS films were performed at room temperature, at an electrical frequency of 2000 Hz and a bias of 100 mV using an E4980a precision inductance, capacitance, and resistance meter (Agilent Technologies, Santa Clara, CA, USA). A custom-built sensor probe station with a programmable *x*-, *y*-, and *z*-axis stage (0.1-μm resolution) enabled the mesoporous Al_2_O_3_-PDMS films to be used to obtain the exact pressure, and a force gauge (Mark-10, Copiague, NY, USA) measured the load.

### 2.4. Fabrication of the Self-Powered Safety Cushion Sensors

We prepared 30 ea, to make a 6 × 5 pixels, of mesoporous Al_2_O_3_-PDMS films attached by conductive graphene tape with a size of 3 × 3 cm^2^ for both the top and bottom side. We put them on the bottom polyethylene terephthalate (PET) film, and the total size was 33 × 27 cm^2^, 6 × 5 pixels, with 3 cm of distance between each cell, then we put the PET film on them. Each cell was connected by a wire to the DAQ 6011, multichannel. It could show the position pressure by color: green was assigned to 1.3–2.5 V, yellow from 2.5–3.8 V, orange from 3.8–5 V, and red over 5 V in real time. 

## 3. Results and Discussion

The schematic diagrams for the synthesis process of the mesoporous Al_2_O_3_-PDMS-based TENGs are shown in Figure 1a. In previous studies, researchers generally made porous structure by using NaCl, ZnO, or polystyrene particles in the PDMS and removing them by immersion in water solution or by chemical treatments. In this study, we used a one-step method without removing anything. The advantages of a one-step method were two-fold. One was a chemical reaction increasing the number and size of pores, given as: (1)$$2\mathrm{Al}+3H_{2}O\;\vec{\mathrm{heat}}{\;\mathrm{Al}}_{2}O_{3}+3H_{2}.$$

Al can easily react with H_2_O because it is an active metal like K, Ca, Na etc., at room temperature or under the drying conditions (80 °C). H_2_ gas, one product of the chemical reaction, increased the volume by increasing the size and the number of pores. Furthermore, Al_2_O_3_ particles had a nano-flake surface following the reaction, increasing the surface area to increase the output performance, as shown in Figure 1c-ii. By most aspects (costs, characteristics like the output performance, particle size), we could choose Al as a suitable active metal and tribo-material. It also has higher density than PDMS and DI water. In a previous study, to increase the output performance, the authors had constructed a multi-layered structure, alternating positively- and negatively-charged layers. In this study, Al, positive material, sank to the bottom because of its relatively higher density than PDMS and DI water. As a result, the top layer was more negatively charged than the bottom layer [8]. The resultant mesoporous Al_2_O_3_-PDMS had a multi-layered structure, and this may have increased the output performance. A photograph of the mesoporous Al_2_O_3_-PDMS (30 × 30 × 7 mm^3^) and the scanning electron microscope image are shown in Figure 1b,c. More Al_2_O_3_ is seen at the bottom (Figure 1c-ii) than at the top (Figure 1c-i).

Figure 1d shows schematically the structure of the mesoporous Al_2_O_3_-PDMS device consisting of a top electrode, mesoporous Al_2_O_3_-PDMS, and the bottom electrode under the vertical contact-separation mode. When the two materials make contact, the top electrode is positively charged and mesoporous Al_2_O_3_-PDMS is negatively charged. In this process, we can separate the two steps, contact and compression. At first, in the contact process, the top electrode firstly contacts the surface of mesoporous Al_2_O_3_-PDMS; it is positively charged, and electrons (e^−^*_sur_*) are moved to the bottom electrode. Then, in the compression process, because of the porous structure of mesoporous Al_2_O_3_-PDMS, the PDMS in the inner surface of pore contacts Al_2_O_3_, and positive (micro-flake Al_2_O_3_) and negative (PDMS) charges increase, additional electrons (e^−^*_inner_*) are moved to the bottom electrode to neutralize the positive charges and the top electrode. Overall, the negative and positive charges occur at not only the surface, but also the inner pores of mesoporous Al_2_O_3_-PDMS, and the output performance is increased. When the external force is removed, firstly, the inner surface and Al_2_O_3_ separate, and e^−^*_inner_* move to the top electrode. Secondly, the top electrode and surface of mesoporous Al_2_O_3_-PDMS separate, and e^−^*_sur_* move to the top electrode to neutralize the negative charges in the bottom electrode; thus, charges occur not only at the surface of mesoporous Al_2_O_3_-PDMS, but also in the inner pore of mesoporous Al_2_O_3_-PDMS, and stacked charges increase. Stacked charges increase due to the advantage of the porous structure (double contact) and can increase the output performance of TENGs. The charge and the voltage were calculated by the following equation:(2)$$\sigma_{P}=\sigma_{sur}+\sigma_{inner}$$
(3)$$V_{OC}=\frac{\left(\sigma_{sur}+\sigma_{inner}\right)x\left(t\right)}{\varepsilon_{0}}$$

Here, $\sigma_{P}$ is the total charge, $\sigma_{sur}$ is the surface charge, $\sigma_{inner}$ is the inner pore charge, *V_OC_* is the open-circuit voltage, and $\varepsilon_{0}$ is the permittivity in a vacuum.

To confirm the additional advantages of Al_2_O_3_ on TENG output performance, we compared the output performance of other materials. We compared the output performance (voltage, current) of metal oxides such as Al, Al_2_O_3_, TiO_2_, SiO_2_, and HfO by sputtering them on an Al sheet (S-Al, S-Al_2_O_3_, S-TiO_2_, S-SiO_2_, S-HfO) and other materials (alkali-treated micro-nano-flaked aluminum hydroxide (Al(OH)_3_), polished Al (P-Al), Al sheet dipped in DI water and dried (Al_2_O_3_)), as shown in Table 1. The results showed that sputtered Al_2_O_3_ had the best output performance (40.96 V, 2.86 μA). In previous studies, materials with greater dielectric constants had better output performance [16]. Although TiO_2_ (*ε_r_* = 80) and HfO (*ε_r_* = 25) have larger dielectric constants than Al_2_O_3_ (*ε_r_* = 9), they had lower output performance than Al_2_O_3_. Based on this result, we believe that Al_2_O_3_ was suitable for fabricating the porous structure and increasing the output performance. 

Figure 2 shows the characteristics of mesoporous Al_2_O_3_-PDMS. First, we confirmed that Al powder changed to Al_2_O_3_ according to the chemical reaction in Equation (1). By energy dispersive spectrometry, we confirmed that the elements Si, Al, C, and O were present. Figure 2a,b shows the SEM and EDS images. Through the SEM image, we confirmed the location of ball-like Al_2_O_3_. By comparison of the SEM images and the distribution of Al (7.48%) and O (28.28%) via EDS images, we see that Al powder, mixed with PDMS and DI water, reacted to form Al_2_O_3_ powder after drying (for 1 h at 80 °C). In previous studies by J. Chun et al., a mesoporous structure made by DI water and demonstrating pores impregnated with Au nanoparticles had 59% of the maximum porosity at a 50% DI water concentration. However, in this study, at the same condition, we got 45.4% of the maximum porosity. When Al MPs were embedded, the porosity increased to approximately 71.35% as the amount of Al increased to 15%, as shown in Figure 2c. The meaning of high porosity is that the mesoporous film is much lighter, more flexible, and deformable. At the same volume, we can get mass ratios of 1:0.37:0.35:0.27 of 0 wt%:5 wt%:10 wt%:15 wt% of Al MP concentration. 

Figure 2d shows the deformability of mesoporous Al_2_O_3_-PDMS (5, 10, 15 wt%) films with a diameter size of 50 mm and a height of 13 mm by ASTM D-695. We limited the jig amplitude to 8 mm and measured the maximum compressed stress at the same conditions. When we compared them, the compressive stress of a mesoporous Al_2_O_3_-PDMS (5 wt%) film was 148.5 MPa with 58.3% of the porosity. However, in the case of mesoporous Al_2_O_3_-PDMS (15 wt%) films, because they had 71.35% of porosity, the maximum compressed stress was the minimum value, 11.7 MPa. That means low compressed stress can be easily deformed, and it is easy to fully contact between the top electrode and a mesoporous Al_2_O_3_-PDMS, like the inset of Figure 2d. We believe that the high porosity and light weight are major factors; if their values increase, they can be higher deformability and can increase the relative capacitance change and output performance of TENGs. 

Figure 2e,f shows that mesoporous Al_2_O_3_-PDMS is much more deformable than non-porous PDMS and mesoporous PDMS (0 wt%). To compare the deformability of non-porous PDMS and mesoporous Al_2_O_3_-PDMS (0, 5, 10, 15 wt%), we measured the relative capacitance with 7 × 7 × 3 mm^2^ films at 0, 16.7, and 33.3 kPa, given respectively as C_0_, C_16.7kPa_, and C_33.3kPa_. When non-porous PDMS, 0, 5, 10, 15 wt% mesoporous Al_2_O_3_-PDMS, is pressed at 33.3 kPa (C_33.3kPa_), the max relative capacitance was 1.08, 1.69, 1.97, 2.07, and 3.03, respectively, at 33.3 kPa. When comparing 0 wt% and 15 wt% of mesoporous Al_2_O_3_-PDMS, the capacitance of the film increased 1.79-times more. Generally, capacitance is expressed by the following Equation (3):(4)$$C=\varepsilon_{0}\varepsilon_{r}\frac{A}{d}.$$

Here, C is the capacitance, $\varepsilon_{0}$ is the permittivity in a vacuum, $\varepsilon_{r}$ is the relative permittivity, *A* is the area of the film, and *d* is the thickness of the film. Based on this equation, if $\varepsilon_{r}$ and *A* are constants, the capacitance inversely depends on the thickness of the film. Thus, as the film is deformable, the film thickness can be easily compressed, and the capacitance increases.

Figure 3 shows the output voltage and output current of non-porous PDMS and P-Al_2_O_3_-PDMS (5, 10, 15 wt%). First, we confirmed the advantage of the porous structure. In Figure 3a, we compare non-porous PDMS and mesoporous P-Al_2_O_3_-PDMS (15 wt%) at 30 N of force, a 4-Hz frequency, a 4-mm gap distance, and a 3 × 3 cm^2^ active area. The output voltage of non-porous PDMS and mesoporous Al_2_O_3_-PDMS (15 wt%) was 39 V and 259.58 V, respectively. The mesoporous structure film had 6.66-times greater output voltage than the non-porous PDMS film. To understand the influence of porous structure in mesoporous Al_2_O_3_-PDMS, we compared one cycle of the output voltage generated by non-porous PDMS and mesoporous Al_2_O_3_-PDMS. In non-porous PDMS, the contact and releasing peaks took 34 and 32 ms to generate electric energy, as shown in Figure 3a-i. However, in mesoporous Al_2_O_3_-PDMS, contact and releasing peaks took 70 and 77 ms, 2.22-times longer, as shown in Figure 3a-ii. The reason is structural. In a previous study, soft membranes and porous structures have been shown to increase the output performance by maximizing effective contact area and the potential difference produced between the Al_2_O_3_ MPs and PDMS inside pores by contact due to different triboelectric tendencies. 

By analyzing the output voltage and fitting the curve of voltage response to dynamic pressure in the range of 5.5–33 kPa, we determined that the voltage pressure sensitivity of the non-porous PDMS and mesoporous Al_2_O_3_-PDMS (5, 10, 15 wt%) reached approximately 0.95, 3.87, 5.57, and 6.71 V/kPa, as shown in Figure 3b. In previous studies, the linearity of pressure sensitivity was divided into several regimes, such as low-pressure and medium-pressure. In this study, however, we found linear pressure sensitivity (sensitivity = 6.37 V/kPa, R^2^ = 0.991 in the 5.5–33 kPa pressure region) across the entire pressure regime. This linearity of the sensitivity of mesoporous Al_2_O_3_-PDMS is affected by the deformability of the film with the porosity (78%) effect; thus, as porosity increases, the linearity of pressure sensitivity (R^2^ value) is high because of the relative capacitance change with thickness change. Like the voltage sensitivity, the current pressure sensitivity reached approximately 0.04, 0.06, 0.10, and 0.18 μA/kPa in Figure 3c, and the linearity was maintained. 

Figure 4 shows the schematic and output voltage and current of gapless TENGs, fabricated by attaching an electrode to both sides of mesoporous Al_2_O_3_-PDMS film. In gapless TENGs, the output performance was much less than contact-separation mode TENGs because the charge density on the surface had a greater influence than in the pores. Thus, by Relation (1), the total charge density (*σ_P_*) was reduced, and the output voltage and current were also reduced. However, the advantages of gapless devices are simple, allowing for large-scale fabrication, low cost, and good special qualities. In other words, gapless TENGs had very poor output performance, but they neither require complicated fabrication, nor limit the large-scale fabrication of practical systems. 

Unlike contact separation mode, charge generation of the mesoporous Al_2_O_3_-PDMS TENG under pressure is caused by the triboelectric effect and electrostatic induction. First, when mesoporous Al_2_O_3_-PDMS film is pressurized, friction occurs between PDMS and nano-flake Al_2_O_3_ MPs in the pores, resulting in negative and positive charges on both surfaces. Furthermore, Al_2_O_3_ with a nano-flake structure increases charge density due to increasing the surface and friction due to electrostatic induction. With these factors, electrons move from the top electrode to the bottom electrode to keep electrical neutrality. When the pressure is released, the surfaces in contact are separated, and electrons flow from the bottom electrode to the top electrode.

Figure 4b–e shows the output voltage, current, and pressure sensitivity of a mesoporous Al_2_O_3_-PDMS (15 wt%) gapless system. As mentioned before, gapless systems had output performance inferior to that of contact-separation systems due to no surface charging, and only inner pore charging. At 33.3 kPa, 4 Hz, and 30 × 30 × 3 mm^3^, we obtained a maximum voltage of 34.00 V and a current of 1.77 μA. Similar to the contact-separation mode, we measured a pressure sensitivity of 1.00 V/kPa and 0.02 μA/kPa.

The output voltage and power of the mesoporous Al_2_O_3_-PDMS (15 wt%) were also measured with external loads varying from 1 MΩ–100 MΩ, as shown in Figure 4f,g. It is clearly seen that the output voltage increased with increasing resistance. Consequently, the instantaneous power of the external resistance was 541.11 μW (60.12 μW/cm^2^) at a resistance of 10 MΩ.

We also demonstrated the self-powered safety cushion sensor for monitoring the sitting position of the human. Conductive tape arrays (6 × 5), total 30 cells, with each cell size of 3 cm × 3 cm on PET films (33 cm × 27 cm), were fabricated on mesoporous Al_2_O_3_-PDMS (15 wt%) films, as shown in Figure 5a. To demonstrate the sitting position as represented by color, green was assigned from 1.3–2.5 V, yellow from 2.5–3.8 V, orange from 3.8–5 V, and red over 5 V in real time. We connected the self-powered safety cushion sensor to the multichannel and monitor, as shown in Figure 5b. To confirm the resolution, we sat at the front, in the middle, and towards the back of the safety cushion sensor, as shown in Figure 5c–e. When sitting at the front, only parts of the hips made contact with the safety cushion sensor and only the front of the safety cushion sensor had output voltage. The closer the hips were to the back of the chair, the greater the proportion of safety cushion sensor output voltage, as shown in Figure 5f–h. We are convinced that the self-powered safety cushion sensor could be applied to wheel chairs to control braking if patients sit too far forward. 

## 4. Conclusions

In summary, we have demonstrated a mesoporous highly-deformable composite polymer with metal oxide by a scalable one-step synthesis unlike the two-step fabrications of previous studies. The oxidation of Al MPs in a polymer film formed a nano-flake morphology on the surface and enhanced the output performance by increasing contact area. As a result, it increased the output performance 6.67-times more than the non-porous PDMS, and the porosity of mesoporous Al_2_O_3_-PDMS (15 wt%) film reached 71.35%. Thus, mesoporous Al_2_O_3_-PDMS (15 wt%) film was found to be more deformable and light weight than the non-porous PDMS film and mesoporous films without metal MPs. As a result, we obtained a relative capacitance change of 3.03, about 1.8-times higher compared with mesoporous films without metal MPs at 33.3 kPa. The output performance of mesoporous Al_2_O_3_-PDMS (15 wt%) films showed an instantaneous voltage of 259.68 V under contact-separation mode, an over six-fold enhancement compared with non-porous PDMS. Further, it had a linear pressure sensitivity of 6.71 V/kPa due to the high porosity. We are convinced that the enhancement was affected by the increased charge density created by contact not only between the top electrode and surface of mesoporous Al_2_O_3_-PDMS, but also between PDMS and the nano-flake structure on metal MPs’ inside pores and high deformation, which changed thickness and increased the electric field. We demonstrated a self-powered safety cushion sensor to detect a human sitting position that can be applied to wheel chair safety. This approach provides a promising large-scale power supply for realizing self-powered safety systems such as matrices, blankets, and road sensors.

## Figures and Tables

**Figure 1 micromachines-09-00656-f001:**
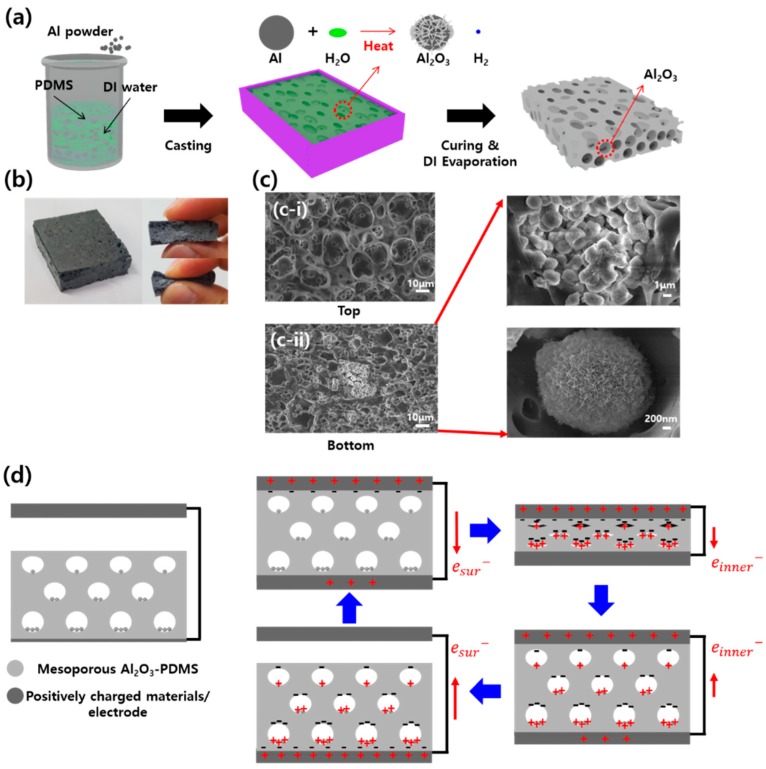
Mesoporous highly-deformable polymer composite and triboelectric nanogenerator (TENG) application. (**a**) Schematic diagram of mesoporous Al_2_O_3_-PDMS fabrication process. (**b**) Photos of highly-deformable mesoporous Al_2_O_3_-PDMS composite polymer. (**c**) The scanning electron microscopy (SEM) images of mesoporous Al_2_O_3_-PDMS top (**c-i**) and bottom (**c-ii**) parts. (**d**) Mechanism of the electricity generation from the mesoporous Al_2_O_3_-PDMS TENG under external force.

**Figure 2 micromachines-09-00656-f002:**
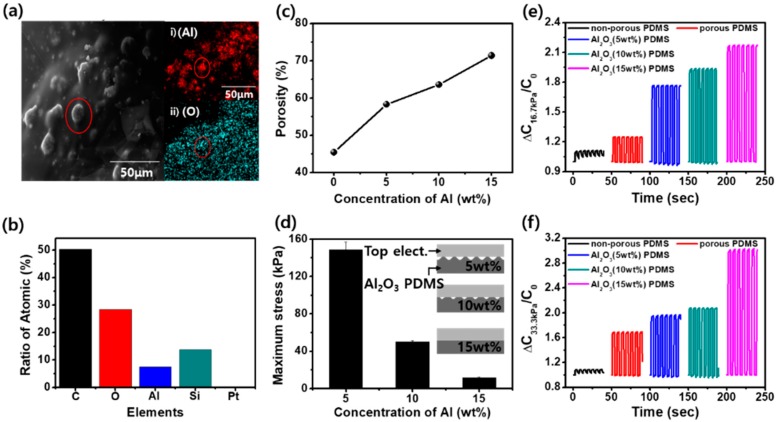
Characterizations of mesoporous Al_2_O_3_-PDMS. (**a**) SEM and EDS images of the elemental maps of (i) Al (red) and (ii) O (cyan); (**b**) ratio of atomic % of mesoporous Al_2_O_3_-PDMS; (**c**) porosity; (**d**) compressed strength at a deformation of 8 mm; the inset is the schematic of the intensity of deformation by compressed strength; and (**e**,**f**) relative capacitance change (ΔC_16.7kPa_, ΔC_33.3kPa_) to the initial capacitance under no pressure (C_0_) of non-porous PDMS and mesoporous Al_2_O_3_-PDMS (0, 5, 10, 15 wt%).

**Figure 3 micromachines-09-00656-f003:**
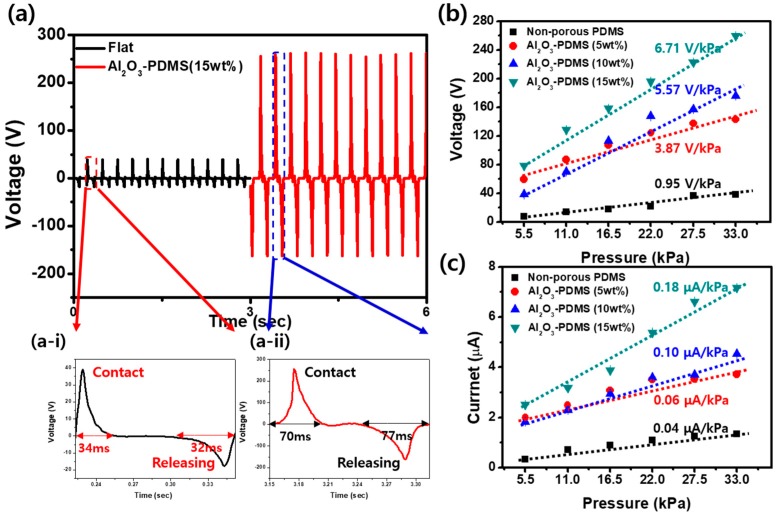
Performance comparison of non-porous PDMS and mesoporous Al_2_O_3_-PDMS (5, 10, 15 wt%). (**a**) The voltage comparison of non-porous PDMS and mesoporous Al_2_O_3_-PDMS at 30 N, 4 Hz and 3 × 3 cm^2^. The insets show a single peak of (**a-i**) PDMS and (**a-ii**) mesoporous Al_2_O_3_-PDMS (15 wt%). (**b**) The voltage and (**c**) current of mesoporous Al_2_O_3_-PDMS (5, 10, 15 wt%) with increasing pressure and the pressure sensitivity.

**Figure 4 micromachines-09-00656-f004:**
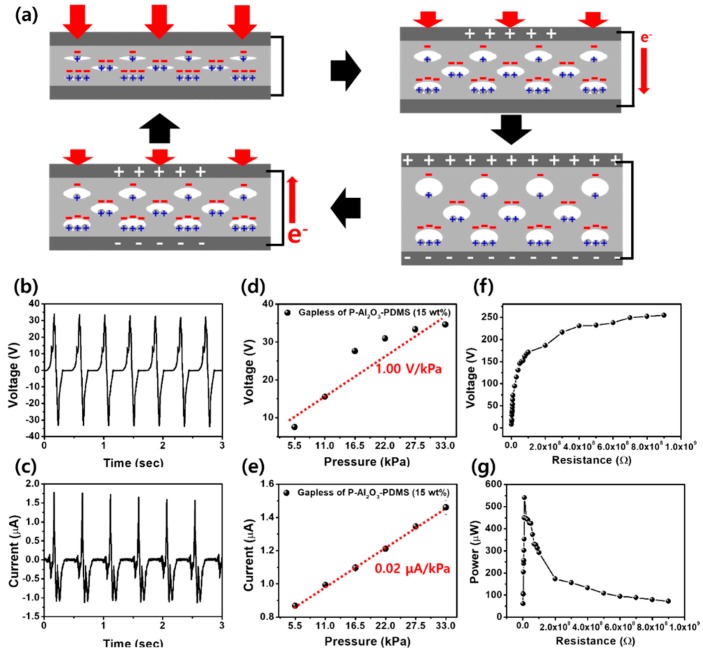
Gapless TENG with a mesoporous Al_2_O_3_-PDMS (15 wt%). (**a**) The schematic of the mesoporous Al_2_O_3_-PDMS (15 wt%) gapless system. (**b**) Output voltage and (**c**) current. The pressure sensitivity of the (**d**) voltage and (**e**) current. (**f**) The output voltage and (**g**) power of mesoporous Al_2_O_3_-PDMS (15 wt%) with resistance of external loads from 10^6^–10^9^ Ω.

**Figure 5 micromachines-09-00656-f005:**
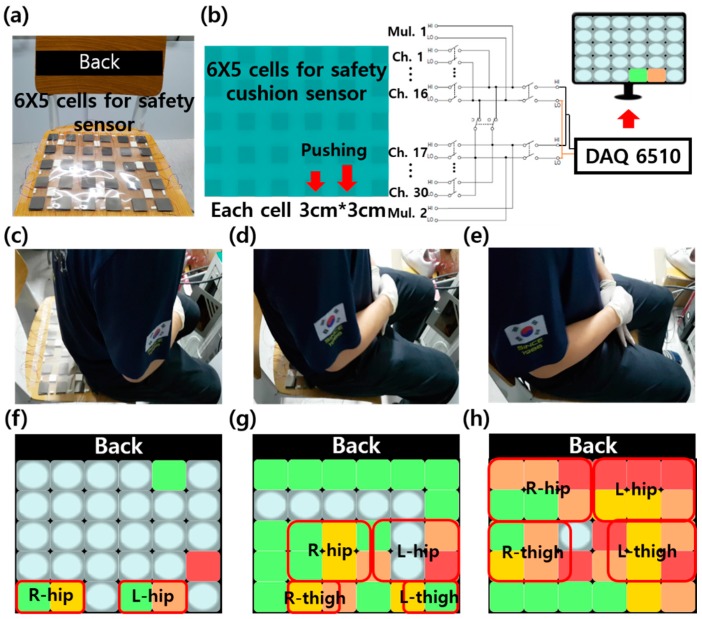
Self-powered safety cushion sensor. (**a**) A photograph of the 6 × 5 cells for the safety sensor, total size of 33 cm × 27 cm; each cell has a size of 3 cm × 3 cm. (**b**) The schematic and circuits of the safety cushion sensor. (**c**–**e**) Detection of human sitting, front, middle, and back of the sensor array. (**f**–**h**) The two-dimensional contour plot via monitoring in real time.

**Table 1 micromachines-09-00656-t001:** Comparison of TENG output performance with various metal oxides and fabrication techniques (sputtering method: S-Al_2_O_3_, S-Al, S-TiO_2_, S-SiO_2_, S-HfO/Alkali treatment: Al(OH)_3_, dipping in water and drying: Al_2_O_3_, polishing: P-Al).

The Output Performance	S-Al_2_O_3_	Al_2_O_3_	S-SiO_2_	S-TiO_2_	S-HfO	S-Al	Al(ref)	P-Al	Al(OH)_3_
Voltage (V)	40.97	36.3	37.30	33.65	31.42	32.29	31.57	36.32	31.42
Current (μA)	2.86	2.44	2.24	2.59	1.92	1.98	1.19	2.82	2.09

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
