# Peer review of "Mesoporous Highly-Deformable Composite Polymer for a Gapless Triboelectric Nanogenerator via a One-Step Metal Oxidation Process"

_micromachines, 2018, doi:10.3390/mi9120656_

Round 1

Reviewer 1 Report

The submitted manuscript investigates a method of making mesoporous PDMS composite material using Al nanoparticles to make Al2O3 pores. The idea is that the mesoporous structure will increase the Triboelectric properties to be used as an energy harvesting device.  The paper is written well and is of interest to readers of the journal who are focused on using triboelectric materials for energy harvesting.

The introduction describes the motivation and reasoning for investigating the issues.

The method section describes the use of PDMS mixed with water and Al MP's to create the mesoporous material. The manuscript described method of making the material. I wonder if the material's properties affect the manufacturability of the material such as does the mesoporous properties affect etching the material or patterning it for MEMS applications.

Results: the authors show 0,5, 10,15 wt% affects, but what happens if you keep increasing to say 20 or 25%, do the voltage and current values start decreasing or does the materials mechanical properties change.  Would be good to show the mechanical properties of the films created as well to see how these are affected.

Page 8 line 267 i would like the authors to give the power density rather than just the power, as a big volume material should harvest more power, so power density is a method of evaluating it compared to other methods.

Author Response

Thank you for comment about this paper.

Q1. The method section describes the use of PDMS mixed with water and Al MP's to create the mesoporous material. The manuscript described method of making the material. I wonder if the material's properties affect the manufacturability of the material such as does the mesoporous properties affect etching the material or patterning it for MEMS applications.

A1. In case of MEMS applications, they need specific control and size. Also, in our study, we fabricated the mesoporous material based on Pdms mixed DI water and Al MPs for TENG applications. We didn't use this material for MEMS. Thank for your comment

Q2. the authors show 0,5, 10,15 wt% affects, but what happens if you keep increasing to say 20 or 25%, do the voltage and current values start decreasing or does the materials mechanical properties change. Would be good to show the mechanical properties of the films created as well to see how these are affected.

A2. Over 15 wt%, Al MPs is not well mixed with PDMS solution and D.I. water. After drying, we can confirm Al MPs is at the bottom of mold, not well mixed with PDMS solution and D.I. water. When comparing volume of 0~30wt% at same mass of PDMS solution and D.I. water, until 15wt%, volume of mesoporous Al2O3 PDMS is increased, however, over that, it is not increased or rather decreased below photo.

Q3. Page 8 line 267 i would like the authors to give the power density rather than just the power, as a big volume material should harvest more power, so power density is a method of evaluating it compared to other methods.

A3. 541.11 μW (60.12 μW/cm2)

Reviewer 2 Report

The manuscript reported a facile one-step approach to synthesize mesoporous Al2O3/PDMS polymers for triboelectric nanogenerators. The highly porous PDMS polymer enhanced the deformation with increased capacitance compared to the nonporous PDMS under mechanical force. As a result, the output performance of the triboelectric device was significantly improved. Porous elastomer composites have shown advantages for triboelectric nanogenerators, the simple and efficient process to fabricate porous elastomer composites is interesting for realization of practical applications. The manuscript is recommended for publication in Micromachines after the comments below are addressed.

1.      The interpretation on the mechanism of the electricity generation process (page 3, line 132) is difficult to be understood, which need to be rephrased. The schematic diagram in Figure 1d may be misleading. An additional schematic image, which shows that the top electrode is releasing from the composite (which causes the current flow to the top electrode) should be added.

The author explains that “At first, top electrode contacts the surface of mesoporous Al2O3-PDMS and it is positively charged and electrons (e-sur) are moved to the bottom electrode.” It is difficult for the reviewer to understand this process. During this step, charge transfer process takes place at the electrode/composite interface to balance the difference in their electronegativity. However, there will be no current flow to the bottom electrode.

According to the schematic in Figure 1d, the electrical field generated by the Al2O3/PDMS interfaced will decrease the electrical field between the top/bottom electrodes, resulting in a reduced output voltage, which will be in contrary to the experimental results.

2.      In the gapless device, the author fabricated the TENGs by attached two electrodes on both sides of the mesoporous Al2O3/PDMS film. The reviewer cannot agree that it is the proper method to fabricate a gapless device. It is difficult to achieve full and constant contact between the electrode and mesoporous film with the attaching approach. Other methods should be considered to avoid the problem, for instance, depositing the electrodes by sputtering a metallic film.  

3.      The English requires more polishing:

“The reasons for studying porous structures are not limited to increasing the output performance, but also apply to life because foam materials apply to fields as diverse as matrix structures, sitting cushions and shoe inserts.”

“Figures 3 (d-e) show that mesoporous Al2O3-PDMS is very more deformable than non-porous
PDMS and mesoporous PDMS (0 wt%)”…

Author Response

Thank you for comment about this paper.

I have some mistakes in  Figure 1. mechanism. So, i fixed electron flow.

1.      The interpretation on the mechanism of the electricity generation process (page 3, line 132) is difficult to be understood, which need to be rephrased. The schematic diagram in Figure 1d may be misleading. An additional schematic image, which shows that the top electrode is releasing from the composite (which causes the current flow to the top electrode) should be added.

The author explains that “At first, top electrode contacts the surface of mesoporous Al2O3-PDMS and it is positively charged and electrons (e-sur) are moved to the bottom electrode.” It is difficult for the reviewer to understand this process. During this step, charge transfer process takes place at the electrode/composite interface to balance the difference in their electronegativity. However, there will be no current flow to the bottom electrode.

According to the schematic in Figure 1d, the electrical field generated by the Al2O3/PDMS interfaced will decrease the electrical field between the top/bottom electrodes, resulting in a reduced output voltage, which will be in contrary to the experimental results.

A. The enhancement of the electrical output performance may be attributed to the relative change in the surface potential level of the PDMS film and the bottom electrode. A potential difference is produced between the Al2O3 MPs and PDMS inside the pores by the contact due to different triboelectric tendencies. This produces a net electric field along the direction from the bottom electrode to the surface of PDMS, that is, the positive charges are in contact with the bottom electrode. This increases the potential difference with the Fermi level of the bottom electrode.

reference.

Chun, J., Kim, J. W., Jung, W.-S., Kang, C.-Y., Kim, S.-W., Wang, Z. L. & Baik, J. M., Mesoporous pores impregnated with Au nanoparticles as effective dielectrics for enhancing triboelectric nanogenerator performance in harsh environments, Energy Environ. Sci., 2015, 8, 3006-3012, [DOI: 10.1039/C5EE01705J]

2.      In the gapless device, the author fabricated the TENGs by attached two electrodes on both sides of the mesoporous Al2O3/PDMS film. The reviewer cannot agree that it is the proper method to fabricate a gapless device. It is difficult to achieve full and constant contact between the electrode and mesoporous film with the attaching approach. Other methods should be considered to avoid the problem, for instance, depositing the electrodes by sputtering a metallic film.  

A. In our study, we fabricate the mesoporous Al2O3 PDMS into rectangular parallelepiped shape by using a molds as shown in figure 1b. Generally, in case of porous film, it is difficult to fully contact and stable between porous film and the electrode. However, our sample has a stable and flat surface without surface pores like a cubic and it is easy to achieve full and constant contact between the electrode and mesoporous film.

Reviewer 3 Report

The authors reported a method for developing gap-free sponge-life triboelectric devices. They use PDMS embedded metal particles and simple oxidation reactions inside the PDMS. The idea is remarkable. Usually, researchers embed oxides in the PDMS and try to increase the dielectric constant of the materials for triboelectric application. This time the particles were post-processed and partially synthesized in the PDMS. Results are remarkable and they also showed a proof of principle application. However, there are minor corrections need to be before publication. Some figures contain very small fonts and it is hard to read for the reader. Please fix it. Figure 6b can be a supplementary figure If you can't make the text larger. 

Author Response

Thank you for comment about this paper.

I fixed it with your comments.

Please check some figures.

Thank you.
